# Cellular Mechanism Underlying Highly-Active or Antiretroviral Therapy-Induced Lipodystrophy: Atazanavir, a Protease Inhibitor, Compromises Adipogenic Conversion of Adipose-Derived Stem/Progenitor Cells through Accelerating ER Stress-Mediated Cell Death in Differentiating Adipocytes

**DOI:** 10.3390/ijms22042114

**Published:** 2021-02-20

**Authors:** Sadanori Akita, Keiji Suzuki, Hiroshi Yoshimoto, Akira Ohtsuru, Akiyoshi Hirano, Shunichi Yamashita

**Affiliations:** 1Department of Plastic Surgery, Wound Repair and Regeneration, School of Medicine, Fukuoka University, 7-45-1 Nanakuma, Jonan-ku, Fukuoka 814-0180, Japan; akitas@hf.rim.or.jp; 2Department of Plastic and Reconstructive Surgery, Nagasaki University Graduate School of Biomedical Sciences, 1-12-4 Sakamoto, Nagasaki 852-8523, Japan; akiyoshi@nagasaki-u.ac.jp (A.H.); hy671117@nagasaki-u.ac.jp (H.Y.); 3Department of Radiation Medical Sciences, Atomic Bomb Disease Institute, Nagasaki University, 1-12-4 Sakamoto, Nagasaki 852-8523, Japan; shun@nagasaki-u.ac.jp; 4Takashi Nagai Memorial International Hibakusha Medical Center, Nagasaki University Graduate School of Biomedical Science, 1-12-4 Nagasaki, Nagasaki852-8523, Japan; ohtsuru@fmu.ac.jp; 5Fukushima Medical University School of Medicine, 1 Hikariga-oka, Fukushima 960-1295, Japan; 6Center for Advanced Radiation Emergency Medicine at the National Institutes for Quantum and Radiological Science and Technology, 4-9-1 Anagawa, Inage-ku, Chiba 263-8555, Japan

**Keywords:** ADSC, HAART, ART, lipodystrophy, adipogenesis, ER stress

## Abstract

Lipodystrophy is a common complication in human immunodeficiency virus (HIV)-infected patients receiving highly active antiretroviral therapy (HAART) or antiretroviral therapy (ART). Previous studies demonstrated that endoplasmic reticulum (ER) stress-mediated unfolded protein response (UPR) is involved in lipodystrophy; however, the detailed mechanism has not been fully described in human adipogenic cell lineage. We utilized adipose tissue-derived stem cells (ADSCs) obtained from human subcutaneous adipose tissue, and atazanavir (ATV), a protease inhibitor (PI), was administered to ADSCs and ADSCs undergoing adipogenic conversion. Marked repression of adipogenic differentiation was observed when ATV was administered during 10 days of ADSC culture in adipogenic differentiation medium. Although ATV had no effect on ADSCs, it significantly induced apoptosis in differentiating adipocytes. ATV treatment also caused the punctate appearance of CCAAT-enhancer-binding (C/EBP) protein homologous protein (CHOP), and altered expression of CHOP and GRP78/Bip, which are the representation of ER stress, only in differentiating adipocytes. Administration of UPR inhibitors restored adipogenic differentiation, indicating that ER stress-mediated UPR was induced in differentiating adipocytes in the presence of ATV. We also observed autophagy, which was potentiated in differentiating adipocytes by ATV treatment. Thus, adipogenic cell atrophy leads to ATV-induced lipodystrophy, which is mediated by ER stress-mediated UPR and accelerated autophagy, both of which would cause adipogenic apoptosis. As our study demonstrated for the first time that ADSCs are unsusceptible to ATV and its deleterious effects are limited to the differentiating adipocytes, responsible target(s) for ATV-induced lipodystrophy may be protease(s) processing adipogenesis-specific protein(s).

## 1. Introduction

Lipodystrophy is the most common adverse event in human immunodeficiency virus (HIV)-infected individuals receiving highly active antiretroviral therapy (HAART) or antiretroviral therapy (ART), and more than half of the patients receiving HAART/ART develop HIV-associated lipodystrophy [1,2,3,4]. Although HAART/ART inhibits virus multiplication and significantly improves the lives of HIV-infected patients, multiple adverse effects, including adipose-tissue damage, have been reported [5,6,7,8].

Previously, one of the two major families of medicine, nucleoside reverse transcriptase inhibitor (NRTI), was reported to cause lipodystrophy through its effects on mitochondria, which results in apoptosis induction in adipocytes [4,9,10,11,12]. Subsequently, another family, protease inhibitor (PI), was found to be associated with lipodystrophy [4,13,14,15]. Although PIs had less effects on mitochondria than NRTIs, they induce ER stress, leading to unfolded protein response (UPR)-dependent apoptosis in primary rat hepatocytes and 3T3-L1 mouse preadipocytes [14,16]. Thus, PIs cause lipodystrophy in many patients receiving HAART/ART [13,15], however, the mechanisms by which PIs induce ER stress in human adipogenic cell lineage have never been determined. In particular, examinations of the effects of PIs on adipose tissue-derived stem cells (ADSCs) are critical, as ADSC-based regenerative medicine could be a promising remedy for lipodystrophy. Thus, it is indispensable for the development of improved HAART/ART for HIV-infected patients.

The ER is an intracellular organelle where protein processing takes place. A number of biochemical and physiological stimuli disturb ER homeostasis, which causes the accumulation of unfolded and misfolded proteins in the ER lumen, leading to UPR [17,18]. There are three major components, ER transmembrane kinase/endoribonuclease inositol requiring enzyme (IRE)1, protein kinase R (PKR)-like ER kinase (PERK), and activating transcription factor (ATF) 6, that transduce ER stress signals [17]. The master regulator, glucose regulatory protein (GRP) 78/binding immunoglobulin protein (BiP), binds to IRE1, PERK, and ATF6, and prevents their activation in unstressed conditions. Upon activation of UPR, GRP78/BiP is released from IRE1, PERK, and ATF6, and they transactivate the expression of the CCAAT-enhancer-binding (C/EBP) protein homologous protein (CHOP) [19], which mediates apoptotic cell death [17,18,20]. As PIs were demonstrated to accumulate unfolded proteins [14,16], ER stress-mediated UPR is considered an essential pathway leading to apoptosis; however, the affected proteins targeted by PIs is still uncertain.

ADSCs have been used as a promising source of stem cells, which can be applied for regenerative medicine [21]. ADSCs can be easily obtained from liposuction aspirates or subcutaneous adipose tissue fragments [22,23]. They are able to expand in vitro with fewer ethical concerns than human embryonic stem cells. Furthermore, accumulating evidence demonstrated that ADSCs harbor the potential for multi-lineage differentiation, including classical mesenchymal lineages, such as adipogenic cells, and non-mesenchymal ectodermal and endodermal lineages [22]. Adipogenic differentiation is a multistep process requiring sequential activation of transcription factors, such as peroxisome proliferator activated receptor-γ (PPAR-γ), in addition to the subsequent expression of adipogenesis-related protein factors [24]. Thus, non-targeted effects of PIs may disrupt adipogenesis, resulting in ER stress. This also raises a possibility that detrimental effects of PIs might be limited to the differentiating adipocytes, which should be determined.

We previously reported that ADSCs from HIV-infected patients exhibit indistinguishable physiological properties and potentials for adipocyte differentiation in vitro [25], which enabled us to examine whether detrimental effects of PIs could have occurred on differentiating human adipocytes. In the present study, we administered a widely used PI, ATV, to human ADSCs and adipocytes differentiated from ADSC in vitro. Our study confirmed for the first time that ATV has no detrimental effects on ADSCs. Molecular analysis revealed that ER stress, limited to differentiating adipocytes, was induced, and ER stress-mediated UPR was involved in apoptosis induction. Thus, the cells of origin in ATV-induced lipodystrophy are not ADSCs, but adipocytes undergoing differentiation, suggesting that protease(s) associated with adipogenesis are the possible target(s) responsible for ATV-induced ER stress.

## 2. Results

### 2.1. Effects of ATV Treatment on Adipogenic Differentiation of ADSCs

ADSCs are able to differentiate to adipocytes in special medium promoting adipogenic differentiation. To examine the effects of ATV on adipogenesis, ADSCs, grown in a serum-free medium until they reached confluence, were then incubated in adipogenic differentiation medium for up to 10 days in the presence of varying doses of ATV. ADSCs started to form multiple tiny lipid droplets within 5 days after changing the medium to a differentiation medium, and the droplets gradually became larger in size by day 10 (Figure 1A, see high power images in Appendix A). ATV treatment at 10 and 20 μM significantly abrogated adipogenic differentiation (Figure 1C,D, see high power images in Appendix A). In order to evaluate the suppressive effects quantitatively, differentiating adipocytes were stained with BODIPY 493/503, a highly sensitive lipophilic fluorescent dye. As shown in Figure 2A (see high power images in Appendix A), multiple lipid droplets were identified in the cytoplasm without ATV, whereas the BODIPY signals were notably reduced by ATV treatment (Figure 2C,D, see high power images in Appendix A). The average BODIPY fluorescence per cell is compared in Figure 2E and we confirmed a significant decrease in lipid droplet formation. Suppression of adipogenesis was also confirmed by Western blot analysis (Figure 3). While ATV treatment to growing ADSCs showed no apparent effect (Figure 3, Appendix A), ATV treatment during adipogenic differentiation significantly reduced the expression of FABP4, adiponectin, and PPARγ, all of which are involved in adipogenesis, whereas they were significantly induced during adipogenesis without ATV (Figure 3, Appendix A).

### 2.2. Differential Effects of ATV on ADSCs and Differentiating Adipocytes

As ATV treatment significantly compromised the adipogenic conversion of ADSCs, we examined whether ATV affects growing ADSCs. As shown in Figure 4, ATV treatment did not induce apoptosis in ADSCs (Figure 4A–D, see high power images in Appendix A), but it significantly induced apoptosis in differentiating adipocytes (Figure 4E–H, see high power images in Appendix A. Separated images are also provided in Appendix A). The results of quantitative analysis are presented in Table 1, confirming that dose-dependent induction of apoptosis is limited to differentiating adipocytes. As shown in Figure 5, cleaved Caspase 3 was only observed in differentiating adipocytes treated with ATV. There were no signs of apoptosis in ADSCs treated with 20 μM ATV (Figure 5, Appendix A).

### 2.3. Induction of ER Stress and Oxidative Stress by ATV Treatment

Based on Western blot analysis, the expression of CHOP and GRP78/BiP proteins in ATV-treated differentiating adipocytes was significantly up-regulated (Figure 5), suggesting that ER stress-mediated UPR is stimulated by ATV treatment. Indeed, the punctate appearance of CHOP was confirmed in differentiating adipocytes treated with 20 μM ATV (Figure 6, see high power images in Appendix A. Separated images are also provided in Appendix A) and its distribution was dependent on the ATV concentration (Appendix A). Involvement of ER stress-mediated UPR was also examined by applying the inhibitors for PERK and IRE-1. Concomitant administration of 40 μM GSK2606414 and 20 μM 4μ8C, which inhibited PERK and IRE-1, respectively, restored adipogenic differentiation and suppressed CHOP expression (Figure 7, Appendix A), and they improved lipid droplet formation, indicating that inhibition of ER stress-mediated UPR reduced ATV-induced suppression of adipogenic conversion of ADSCs (Figure 8).

The Western blot analysis also revealed that autophagy is up-regulated in ATV-treated differentiating adipocytes. Of note, adipogenesis alone did not stimulate ER stress according to the CHOP and GRP/BiP expression, but it accelerated autophagy because the ATG12–ATG5 complex was formed upon adipogenic conversion (Figure 5, Appendix A).

As ROS-mediated cell death was reported after some of the PIs’ treatments, ATV treatment-associated oxidative stress was measured using 3’-(*p*-aminophenyl)-fluorescein (APF), dihydroethidium (DHE), and MitoSox-red, which predominantly detect hydroxy radicals, intracellular superoxide, and mitochondrial superoxide, respectively. As shown in Figure 9 (see high power images in Appendix A), an increased number of MitoSox-positive cells was observed, and an altered level of MitoSox red fluorescence was detected in differentiating adipocytes treated with 20 μM ATV (Figure 9B, separated images are provided in Appendix A); however, its change was not significant (Figure 9E). Similarly, oxidative stress levels according to DHE and APF did not significantly increase after ATV treatment (Figure 9E). On conclusion, we were unable to detect an increased level of DNA damage.

## 3. Discussion

Subcutaneous adipose-tissue damage, which is a cause of lipodystrophy, is common among those receiving HAART/ART [1,2,3,4,5,6,7,8]. Similar to earlier observations with the usage of NRTIs, PIs were reported to cause lipodystrophy [4,13,14,15]. Several in vitro studies using murine cell culture systems demonstrated that PIs induce apoptosis in preadipocytes, which resulted in defective adipogenesis [14,16]. ER stress is involved in impaired adipogenesis; however, the mechanism underlying this adverse effect has yet to be determined in human adipogenic cell lineage. In particular, cell context-dependent effects have yet to be examined; therefore, we used human ADSC and ADSC-derived adipocytes to compare the detrimental effects of ATV.

Among PIs, ATV was reported to have milder effects than other PIs, such as ritonavir [26,27]; however, the present study confirmed that ATV attenuates adipogenic conversion of human ADSCs. There could be two possibilities involved in ATV-induced adipogenesis suppression. One of them is suppression of the initiation of adipogenic differentiation, while another is inhibition of lipid droplet formation. According to the result shown in Figure 3, ATV treatment concurrently with the addition of differentiation medium significantly decreases the level of PPARg, which is a factor involved in adipogenic differentiation. Therefore, it is highly likely that ATV treatment abrogates adipogenic differentiation. Although the current in vitro culture system could not mimic the in vivo situation, our results suggest no neoformation of adipocytes in subcutaneous adipose tissue in patients, which needs to be determined in future experiments.

As proteins specific for adipogenic differentiation were not expressed in ADSCs (Figure 3), protease(s) responsible for the ATV-mediated effect were likely to be involved in processing of the differentiation-related proteins. As it was possible that ATV reacted with components in the differentiation medium caused ER stress, we used exponentially growing ADSCs that did not differentiate even in a differentiation medium, confirming that ER stress was induced only in the differentiating adipocytes in the presence of ATV.

We confirmed that ATV induced ER stress, as described previously in different cell systems [28,29], but ER stress was not observed in human ADSCs treated with ATV (Figure 5), suggesting that the adverse effects of ATV are limited to differentiating adipocytes. Furthermore, adipogenesis alone was not an inducer of ER stress, whereas adipogenic differentiation negated ER stress because we were unable to detect CHOP expression and noted reduced expression of GRP78/BiP in differentiating adipocytes (Figure 5). Adipogenesis and ER stress are known to be related, and adipogenesis stimulated ER stress-mediated UPR [30,31,32,33]. Although there was an exception, most studies reported that it compromises adipogenic differentiation [30,31,32]. Although these studies used multiple cell systems, the majority used mouse 3T3-L1 cells, which exhibited ER stress-mediated UPR during adipogenesis [30,31,34]. In contrast, our study using human ADSCs differentiated in vitro demonstrated that adipogenic differentiation did not cause ER stress. Our conclusion was also supported by the lack of UPR-mediated apoptosis induced during adipogenesis (Figure 4). Thus, it is reasonable to conclude that we had better use human adipogenic stem cells, ADSCs, and their differentiated progenies for a better understanding of the adverse effects of PIs.

While we did not detect ER stress-mediated UPR during adipogenesis, ATV treatment significantly induced UPR because CHOP and GRP78/BiP expression significantly increased. This was also confirmed by the punctate CHOP signals (Figure 6) and marked apoptosis induction, not necrosis (Figure 4 and Figure 5). Considering that the cells need to process great amount of proteins during adipogenesis, it is possible that accumulation of unprocessed proteins in the presence of ATV gives rise to the ER stress. In fact, the experiments using ER stress-mediated UPR inhibitors, GSK2606414 and 4μ8C, targeting PERK and IRE-1, respectively, demonstrated that they alleviated ATV-mediated suppression of adipogenic conversion of ADSCs (Figure 8). Thus, considering that ATV treatment did not induce apoptosis in ADSCs, it can be concluded that its adverse effects are limited to adipogenic cells undergoing adipogenesis through activation of UPR. This suggests that adipogenesis-induced protein synthesis is a prerequisite for ER stress induction by ATV treatment. Indeed, adipogenesis is a process in which there is marked change in the transcription profile. HIV-infection and HAART were reported to alter this profile [35,36,37]. Accordingly, many proteins, particularly several secreted factors, are newly synthesized during adipogenesis. As the ER is where these secreted proteins are processed, it is possible that protease(s) associated with protein processing are inhibited by PIs, which induces ER stress-mediated UPR. This possibility should be assessed in future studies, as it may aid in the development of treatments mitigating lipodystrophy in HIV-infected patients receiving HAART/ART.

The current study also provided evidence of autophagy induction during the adipogenic conversion of ADSCs. Autophagy has been described as another physiological process activated during adipogenesis [38,39,40]. Indeed, autophagy is indispensable for white adipose tissue development [41]. As the ADSCs used in this study were obtained from subcutaneous adipose tissue, adipocytes differentiated from ADSCs may be similar to those derived from white adipose tissue. As shown in Figure 5, adipogenic differentiation stimulates ATG12–ATG5 conjugation, which is an essential step towards autophagy initiation. However, the expression level of LC3B-I was down-regulated during adipogenesis. As shown in Figure 3, the levels of β-actin were not changed, thus the LC3B-I/β-actin ratio significantly decreased. In contrast, LC3B-II, which is the activated form of LC3B-I by conjugation with phosphatidylethanolamine, did not decrease. Therefore, we were able to conclude that phagophore elongation and autophagosome formation were stimulated during adipogenic conversion of ADSCs, although autophagosome degradation was accelerated.

Our another novel finding was that ATV treatment accelerated autophagy in differentiating adipocytes, whereas it had no effect on ADSCs (Figure 5). Although its physiological significance is not known, autophagic cell death may be another reason that ATV treatment abrogates adipogenic conversion of ADSCs. ER stress-mediated UPR and autophagy are known to be related [42,43], and ATF4-dependent SQSTM1 expression is indeed involved in executing selective autophagy in the ER [43]. ER stress-mediated UPR also accelerates mitophagy [44], which is another route compromising adipogenesis. In fact, we observed an increased level of mitochondrial damage, as MitoSox-red fluorescence was increased. Although it was not a level causing DNA damage and cell death, targeting autophagy may provide another method to alleviate lipodystrophy; however, target(s) that induce ER stress-mediated UPR in the presence of ATV should be identified.

In summary, we demonstrated for the first time that adipogenic cell atrophy leads to ATV-induced lipodystrophy, which is mediated by ER stress-induced UPR in human adipogenic cell lineage, which causes apoptosis in differentiating adipocytes, as well as by autophagy, whereas oxidative stress played no role. The deleterious effects of ATV were obviously cell context-dependent, and there was no apoptosis or autophagy induced in ADSCs treated with ATV, suggesting that drugs targeting responsible protease(s) in differentiating adipocytes may be applicable for alleviating the adverse effects of atazanavir treatment.

## 4. Materials and Methods

### 4.1. ADSCs and Culture

Human ADSCs were obtained from commercially available sources (Lonza, Tokyo, Japan). ADSCs were plated onto collagen type-I-coated plastic culture flasks (BD Bioscience, Tokyo, Japan) in serum-free medium for primate embryonic stem cells (Primate ES medium, ReproCELL, Tokyo, Japan). ADSCs were subcultured as described previously [44]. Medium was changed every 3 days to maintain exponentially growth.

### 4.2. Adipocyte Differentiation and ATV Treatment

ADSCs (1 x 10^5^) were plated onto 22 mm × 22 mm type I collagen-coated cover glass slips and cultured in serum-free medium until they reached confluence. Then, the culture medium was changed to differentiation medium (DM-2, ZenBio, Inc., Durham, NC), and cells were cultured for 7 or 10 days until they were fixed with 4% formalin. Atazanavir sulfate (ATV) was kindly provided by Bristol-Myers Squibb K. (Tokyo, Japan). ATV was dissolved in DMSO at 20 mM and stored at −30 °C until usage. ATV was added concurrently when medium was changed to differentiation medium, and cells were treated with ATV for 7 or 10 days until they were fixed with 4% formalin. Differentiation medium with or without ATV was changed every 3 days. ER stress inhibitors, GSK2606414 and 4μ8C (Selleck, Co., Tokyo, Japan), were dissolved in DMSO at 20 mM and stored at −30 °C until usage. The final concentration was 40 μM for GSK2606414 and 20 μM for 4μ8C, and they were added concurrently with ATV when medium was changed to differentiation medium. Cells were treated with ATV and inhibitor for 10 days until they were fixed with 4% formalin.

The fixed cells were stained with 10 µg/mL of BODIPY 493/503 (D-3922, Invitrogen, Carlsbad, CA) for 20 min at room temperature and the nuclei were counterstained with 0.1 μg/mL of DAPI. Lipid droplets were examined under a fluorescence microscope (F3000B, Leica, Tokyo, Japan). Digital images were captured and the images were analyzed using FW4000 software (Leica). Cells containing multiple lipid droplets in more than 50% of the cytoplasm were counted as differentiation-positive cells. In order to quantify the average fluorescence per cell, areas were randomly selected and the sum of the pixel intensity within the marked area was calculated using FW4000 software. Total green fluorescence was divided by the total blue fluorescence of DAPI staining to calculate the relative fluorescence.

### 4.3. Immunofluorescence

Cells (5 × 10^4^) were replated onto collagen type-I-coated 22 mm × 22 mm coverslips. The cells were fixed with 4% formaldehyde for 10 min on ice, followed by the permeabilization with 0.05% Triton X-100 for 5 min on ice. After extensive washing with PBS, primary antibodies diluted in TBS-DT (20 mM Tris-HCl, pH7.6, 137 mM NaCl, 0.1% Tween 20, 125 µg/mL ampicillin, and 5% skim milk) were applied for 2 h at 37 °C, followed by the Alexa Fluor-labeled secondary antibodies for 1 h at 37 °C. Nuclei were counterstained with 1 μg/mL of DAPI. The antibodies used in this study were anti-phosphorylated histone H2AX at serine 139 (clone 2F3, BioLegend, San Diego, CA, USA), anti-53BP1 (A300-272A, BioLegend), anti-CHOP (L63F7, CST Tokyo, Japan), Alexa Fluor 555-labeled anti-mouse IgG (A21422, Thermo Fisher Scientific, Tokyo, Japan), and Alexa Fluor 488 or 555-labed anti-rabbit IgG (A11008 or A21428, Thermo Fisher Scientific). Images were captured with a fluorescence microscope (DM6000B, Leica) and analyzed using FW4000 software (Leica).

### 4.4. DNA Damage and Apoptosis Detection

DNA double-strand breaks and apoptosis were detected by immunofluorescence using antibodies against phosphorylated histone H2AX at serine 139 (clone 2F3, BioLegend), and anti-53BP1 (A300-272A, BioLegend). As phosphorylated histone H2AX and 53BP1 form discrete foci at the sites of DNA double-strand breaks, DNA damage induction was evaluated by detecting foci. In cells undergoing apoptosis, DNA is fragmented; therefore, multiple DNA double-strand breaks are induced, which are detectable by homogeneous nuclear staining with an antibody against phosphorylated histone H2AX at serine 139, not with 53BP1 antibody. The primary antibodies were detected by Alexa Fluor 555-labed anti-mouse IgG (A21422, Thermo Fisher Scientific). Nuclei were counterstained with 1 μg/mL of DAPI. Images were captured with a fluorescence microscope (DM6000B, Leica) and analyzed using FW4000 software (Leica). The percentage of apoptotic cells was calculated by dividing the number of cells with heavily phosphorylated histone H2AX signals by the number of total cells determined by the number of DAPI-positive nuclei. The percentage of cells with DNA double strand breaks was calculated by dividing the number of cells with 53BP1 foci by the number of total cells determined by the number of DAPI-positive nuclei.

### 4.5. Western Blotting

For protein preparation, ADSCs were plated onto collagen type-I-coated plastic culture flasks (BD Bioscience Tokyo, Japan) in serum-free medium for primate embryonic stem cells (Primate ES medium, ReproCELL, Tokyo, Japan) with or without ATV. ADSCs plated onto collagen type-I-coated plastic culture flasks were cultured until they reached confluence, then the culture medium was changed to differentiation medium (DM-2, ZenBio, Inc., Durham, NC, USA) and cells were cultured for 7 or 10 days with or without ATV. Cells were lysed in lysis buffer (50 mM Tris-HCl (pH 7.2), 150 mM NaCl, 1% NP-40, 1% sodium deoxycholate, and 0.1% SDS) containing 1 mM 4-(2-aminoethyl)-benzensulfonyl fluoride hydrochloride. The cell lysate was cleared by centrifugation at 15,000 rpm for 10 min at 4 °C and the supernatant was used as total cellular protein. The total protein concentration was measured by the BCA protein assay (Pierce, Rockford, IL, USA). Protein samples (8 or 16 µg) were electrophoresed on SDS-polyacrylamide gel and electrophoretically transferred to a polyvinyl difluoride membrane in transfer buffer (100 mM Tris and 192 mM glycine). After blocking with 10% skim milk, the membrane was incubated with the primary antibodies, followed by alkaline-conjugated anti-mouse or anti-rabbit IgG antibodies (Promega Tokyo, Japan). The bands were visualized after addition of nitroblue tetrazolium/5-bromo-4-chloro-3-indolyl phosphate as a substrate. The primary antibodies used in this study were anti-adiponectin (clone 19F1, Abcam Co. Ltd., Tokyo, Japan), anti-FABP4 (Abcam Co. Ltd., Tokyo, Japan), anti-PPARγ (clone 81B8, Cell Signaling technology Japan), and anti-β-actin (Cell Signaling technology Japan). The density of each band was analyzed using Image J software. The intensity of the bands obtained by anti-β-actin antibody was used to evaluate protein loading, so we checked the band intensity of β-actin within the same series of proteins samples several times.

### 4.6. Measurement of ROS

The intracellular oxidative stress level was measured by 3’-(*p*-aminophenyl)-fluorescein (APF) (Wako, Osaka, Japan) and dihydroethidium (DHE) (Wako). Cells cultured in T25 flasks were washed with phosphate-buffered saline (PBS) and treated with 5 μM APF or 5 μM DHE in PBS for 60 min at 37 °C in a 5% CO_2_ incubator. After treatment, the cells were suspended in PBS at 4 × 10^4^ cells/mL, and green and red fluorescence intensity was measured by a fluorometer (JASCO, Tokyo, Japan). The excitation and emission wavelengths were 490 nm and 515 nm for APF, and 500 nm and 580 nm for DHE, respectively.

Mitochondrial damage was quantified by MitoSox-Red. Cells cultured in T25 flasks were washed with PBS and treated with 1 μM MitoSox-Red (Invitrogen) in PBS for 20 min at 37 °C in a 5% CO_2_ incubator. After treatment, the cells were trypsinized, suspended in PBS at 4 × 10^4^ cells/mL, and red fluorescence intensity was measured by a fluorometer (JASCO, Tokyo). The excitation and emission wavelengths were 500 nm and 580 nm, respectively. The nuclei were counterstained with 0.1 μg/mL of DAPI. Relative fluorescence was calculated by dividing total green or red fluorescence by the total blue fluorescence of DAPI staining.

### 4.7. Data Analysis

All experiments were repeated at least three times. The data are expressed as the mean ± SD. The Wilcoxon rank test was used to evaluate significant differences between two groups, and two-tailed Student’s *t*-test was used to compare the band intensities obtained from Western blotting analyses. *p*-values of less than 0.05 were considered significant.

## Figures and Tables

**Figure 1 ijms-22-02114-f001:**
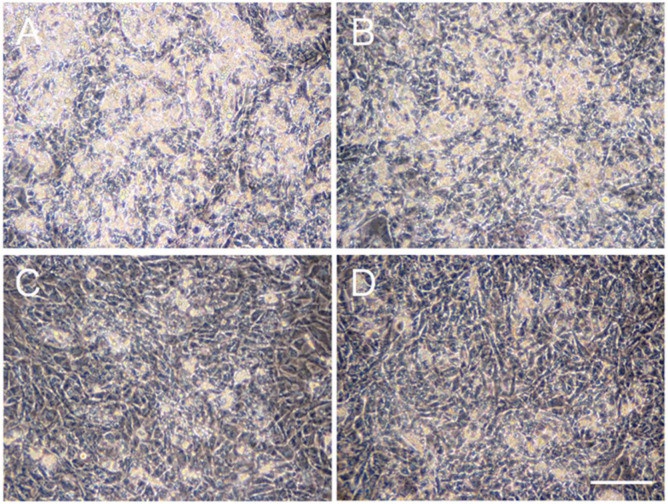
Suppression of adipogenic differentiation of adipose tissue-derived stem cells (ADSCs) by atazanavir (ATV). ADSCs, cultured in serum-free medium until they reached confluence, were incubated in adipogenic differentiation medium for 10 days with varying concentrations of ATV. (**A**) control, (**B**) 5 μM ATV, (**C**) 10 μM ATV, and (**D**) 20 μM ATV. Images were taken under a phase-contrast microscope. White scale bar indicates 250 μm.

**Figure 2 ijms-22-02114-f002:**
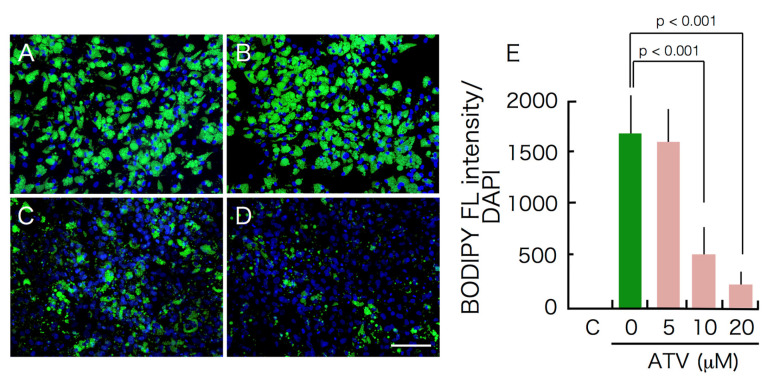
Suppression of adipogenesis-dependent accumulation of lipid droplets by atazanavir (ATV). Adipose tissue-derived stem cells (ADSCs), cultured in serum-free medium until they reached confluence, were incubated in adipogenic differentiation medium for 10 days with varying concentrations of ATV. (**A**) control, (**B**) 5 μM ATV, (**C**) 10 μM ATV, and (**D**) 20 μM ATV. Cells were fixed with 4% formalin and stained with BODIPY 493/503. Images were taken under a fluorescence microscope. White scale bar indicates 250 μm. (**E**) Fluorescence images were taken under a fluorescence microscope and green fluorescence (FL) of BODIPY was divided by the blue fluorescence of DAPI. C, control. Experiments were repeated three times.

**Figure 3 ijms-22-02114-f003:**
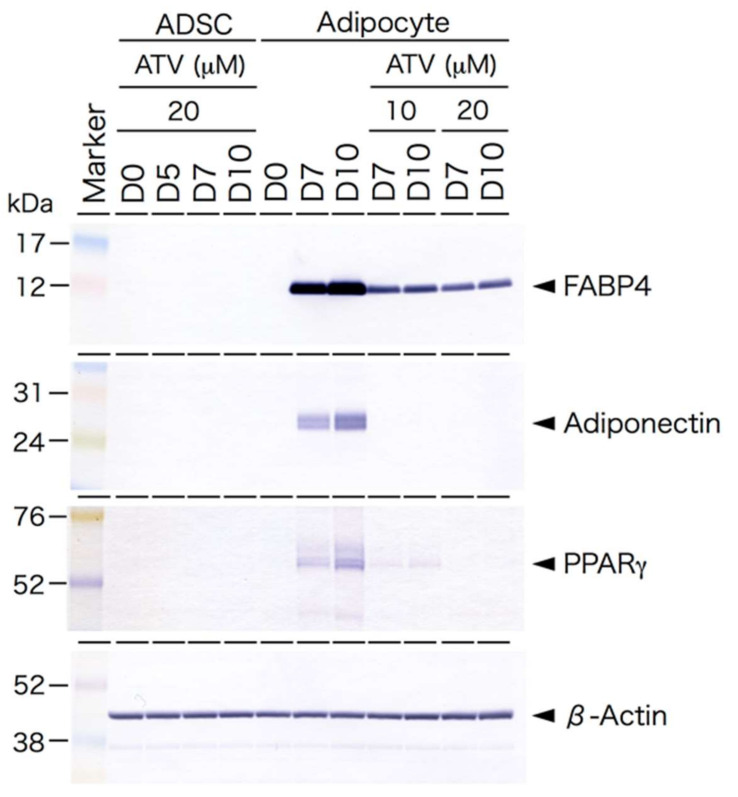
Western blot analysis of adipogenesis-related proteins ADSCs cultured in a serum-free medium, treated with 20 μM ATV for the indicated time (ADSC). D0 indicates the day when ATV was administrated. ADSCs, cultured in serum-free medium until they reached confluence, were cultured in adipogenic differentiation medium for the indicated time with or without ATV at 10 μM or 20 μM for the indicated time (adipocyte). D0 indicates the day when the medium was changed to differentiation medium and ATV was concurrently added. Protein samples (8 or 16 µg) were electrophoresed on SDS-polyacrylamide gel, electrophoretically transferred to a polyvinyl difluoride membrane, and the membrane was incubated with the indicated primary antibodies. PPARγ, peroxisome proliferator activated receptor-γ.

**Figure 4 ijms-22-02114-f004:**
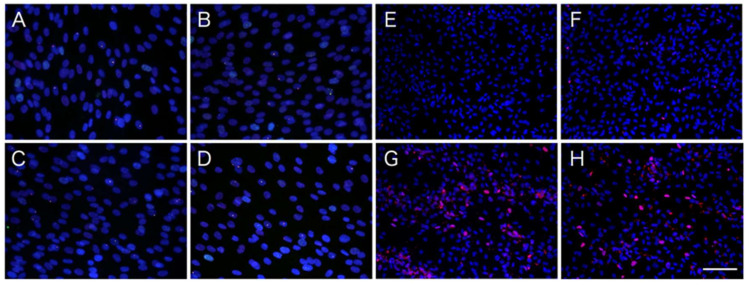
DNA damage and apoptosis induction in ADSCs and during adipogenesis by ATV treatment. (**A**–**D**) ADSCs, cultured in a serum-free medium, were treated with varying concentrations of ATV for 7 days. (**E**–**H**) ADSCs, cultured in serum-free medium until they reached confluence, were incubated in adipogenic differentiation medium for 10 days concurrently with varying concentrations of ATV. (**A**, **E**) control, (**B**, **F**) 5 μM ATV, (**C**, **G**) 10 μM ATV, and (**D**, **H**) 20 μM ATV. Images were taken under a fluorescence microscope. Red fluorescence indicates phosphorylated histone H2AX. White scale bar indicates 125 μm.

**Figure 5 ijms-22-02114-f005:**
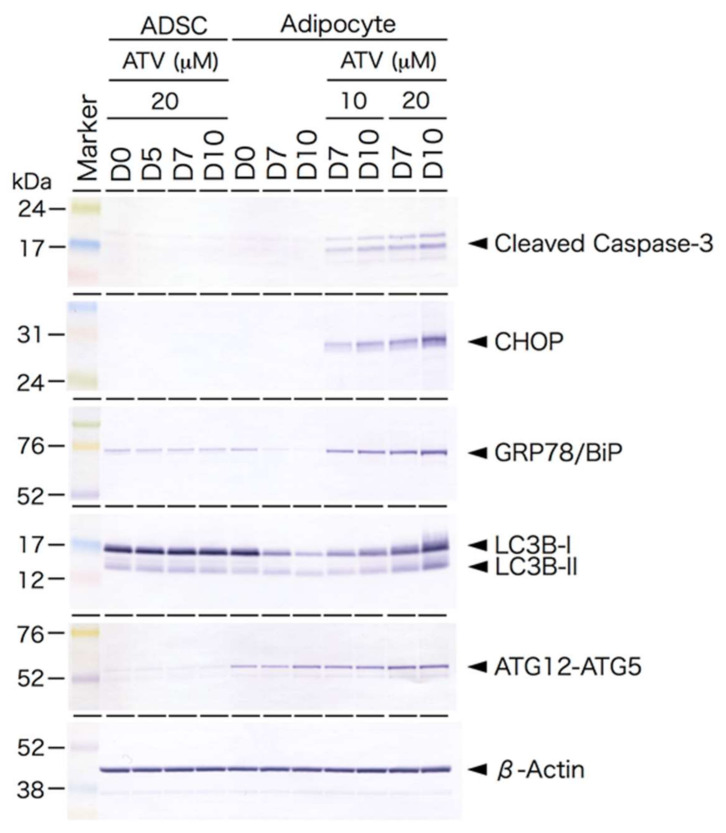
Western blot analysis of endoplasmic reticulum (ER) stress-mediated unfolded protein response (UPR) and autophagy-related proteins ADSCs cultured in a serum free-medium, treated with 20 μM ATV for the indicated time (ADSC). D0 indicates the day when ATV was administrated. ADSCs, cultured in serum-free medium until they reached confluence, were cultured in adipogenic differentiation medium for the indicated time with or without ATV at 10 μM or 20 μM for the indicated time (Adipocyte). D0 indicates the day when the medium was changed to differentiation medium and ATV was concurrently added. Protein samples (8 or 16 µg) were electrophoresed on SDS-polyacrylamide gel, electrophoretically transferred to a polyvinyl difluoride membrane, and the membrane was incubated with the indicated primary antibodies. CHOP, CCAAT-enhancer-binding (C/EBP) protein homologous protein.

**Figure 6 ijms-22-02114-f006:**
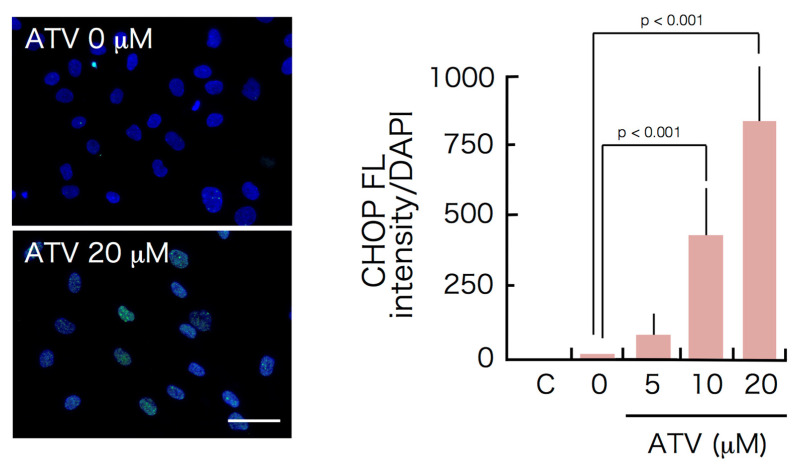
Induction of discrete foci of CHOP during adipogenesis of ADSCs treated with ATV. ADSCs, cultured in a serum-free medium until they reached confluence, were incubated in adipogenic differentiation medium for 7 days with varying concentrations of ATV. Images were taken under a fluorescence microscope. Green punctate fluorescence signals indicate CHOP. Green fluorescence was divided by blue fluorescence to calculate CHOP expression/cell. C, control. White scale bar indicates 100 μm. The experiments were repeated three times.

**Figure 7 ijms-22-02114-f007:**
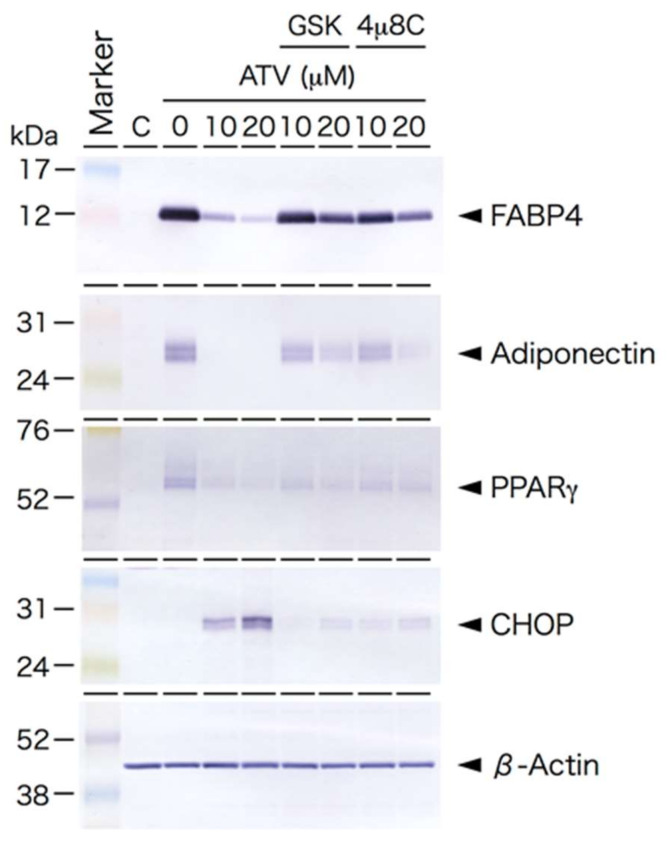
Effects of ER stress inhibitors on protein expression. GSK2606414 and 4μ8C, which inhibit protein kinase R (PKR)-like ER kinase (PERK) and inositol requiring enzyme 1 (IRE-1), respectively, were dissolved in DMSO at 20 mM and stored at −30 °C until usage. The final concentration was 40 μM for GSK2606414 and 20 μM for 4μ8C, and they were added concurrently with ATV when medium was changed to differentiation medium. Cells were treated with ATV and inhibitor for 10 days before protein extraction. Protein samples (8 or 16 µg) were electrophoresed on SDS-polyacrylamide gel, electrophoretically transferred to a polyvinyl difluoride membrane, and the membrane was incubated with the indicated primary antibodies.

**Figure 8 ijms-22-02114-f008:**
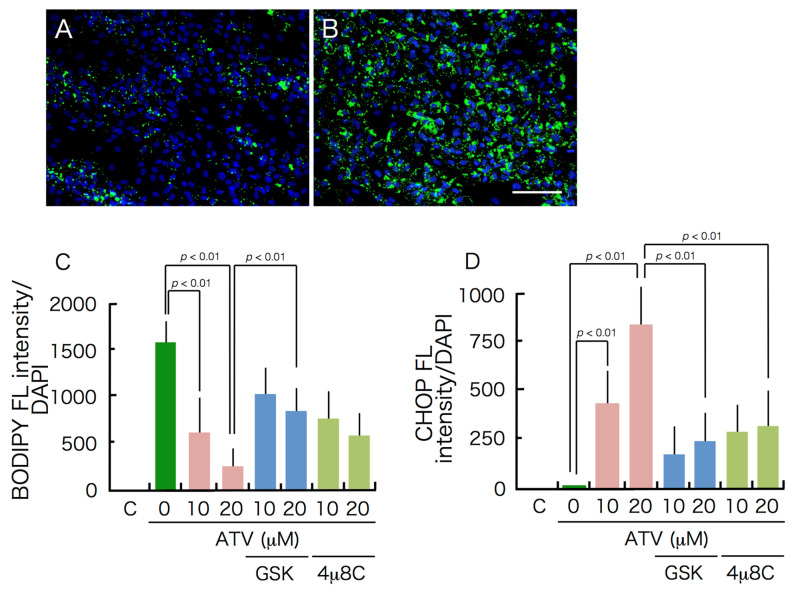
Effects of ER stress inhibitors on differentiation. GSK2606414 and 4μ8C, which inhibit PERK and IRE-1, respectively, were dissolved in DMSM at 20 mM and stored at −30 °C until usage. The final concentration was 40 μM for GSK2606414 and 20 μM for 4μ8C, and they were added concurrently with ATV when medium was changed to differentiation medium. Cells were treated with ATV and inhibitor for 10 days until they were fixed with 4% formalin. (**A**, **B**) BODIPY 493/503 staining. (**A**), ADSCs incubated in adipogenic differentiation medium for 10 days with 20 μM ATV, (**B**) ADSCs incubated in adipogenic differentiation medium for 10 days with 20 μM ATV and 40 μM GSK2606414, (**C**) the green fluorescence of BODIPY was divided by the blue fluorescence of DAPI, and (**D**) cells were stained with CHOP antibody and the green fluorescence of CHOP was divided by the blue fluorescence of DAPI.

**Figure 9 ijms-22-02114-f009:**
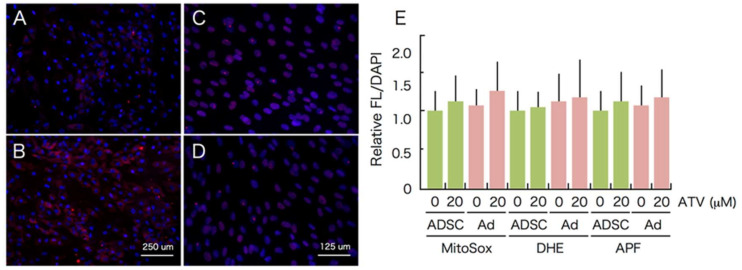
ROS and DNA damage induction during adipogenesis in the presence of ATV. ADSCs, cultured in a serum free medium until they reached confluence, were incubated in adipogenic differentiation medium for 7 days with 20 μM ATV. After fixation, cells were incubated with MitoSox-red (**A**, **B**) or stained with anti-53BP1 antibody (**C**, **D**). (**A**, **C**) control, (**C**, **D**) 20 μM ATV. (**E**) ADSCs were incubated in adipogenic differentiation medium for 7 days with 20 μM ATV. After fixation, cells were incubated with MitoSox-red, DHE, and APF. Relative fluorescence was measured as described in Materials and Methods.

**Table 1 ijms-22-02114-t001:** Apoptosis in differentiating adipocytes by ATV treatment.

Cells.	ATV Dose	% Apoptotic Cells	*p*-Value *
ADSC		0.0	NS
	0 μM	0.0	NS
	5 μM	0.0	NS
	10 μM	0.0	NS
	20 μM	0.0	NS
Adipocytes			
	0 μM	0.52 ± 0.09	NS
	5 μM	0.97 ± 0.10	NS
	10 μM	2.52 ± 0.19	*p* < 0.01
	20 μM	4.34 ± 0.25	*p* < 0.01

* Statistical significance to adipocyte control. NS: not significant. ADSCs (adipose tissue-derived stem cells), cultured in a serum-free medium until they reached confluence, were treated with varying concentration of ATV (atazanavir) for 7 days. For adipocyte experiments, ADSCs, cultured in a serum-free medium until they reached confluence, were incubated in an adipogenic differentiation medium for 7 days concurrently with varying concentration of ATV. Percentage of apoptotic cell were calculated by dividing the number of cells with heavily phosphorylated histone H2AX signals (homogeneous signals) by the number of total cells measured by the number of DAPI positive nuclei.

## Data Availability

Data sharing not applicable.

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
