# Peer review of "Cellular Mechanism Underlying Highly-Active or Antiretroviral Therapy-Induced Lipodystrophy: Atazanavir, a Protease Inhibitor, Compromises Adipogenic Conversion of Adipose-Derived Stem/Progenitor Cells through Accelerating ER Stress-Mediated Cell Death in Differentiating Adipocytes"

_ijms, 2021, doi:10.3390/ijms22042114_

Round 1
Reviewer 1 Report
In the present work the authors show that atazanavir abrogated adipogenic differentiation of Human Stem cells involving apoptosis, autophagy, and ER stress induction.
It is a very interesting study; however, I have some doubts about whether the study design is incomplete.
In adipose tissue, the adipogenesis process is a process that occurs during the growth. As I see in WB and other figures, differentiation process is aborted, since there is only the initiation phase, characterized by the increase of FABP4, while maturation is almost absent since PPARg is non-existent according to WB. According to figure 2 with DAPI, with 10 and 20uM of ATV adipogenesis is almost non-existent, perhaps the low signal of immunostaining that the authors detected is unspecific. This suspicion is confirmed with the WB where neither PPARg nor adiponectin are present. Thus, according to the results obtained on HMSCs, what would occur in adipose tissue of the patients treated with ATV, is that there is no neoformation of adipocytes in subcutaneous adipose tissue.
-What is the effect of ATV on the mature adipocytes within the subcutaneous adipose tissue of these patients?
-My question is; whether the lipodystrophy detected with ATV treatment is, may be, not due to the inhibition of adipocytes neoformation, but rather to an imbalance between this process and the potential necrosis effects of ATV on mature adipocytes within the adipose tissue?.
At least the authors should discuss this issue
-In WB panels, it seems that the authors do not represent results from the same Blot as control profile seems to be different of that of the other proteins, for example; comparing beta-actin and FABP4 in figure 3.
Author Response
In the present work the authors show that atazanavir abrogated adipogenic differentiation of Human Stem cells involving apoptosis, autophagy, and ER stress induction.
It is a very interesting study; however, I have some doubts about whether the study design is incomplete.
In adipose tissue, the adipogenesis process is a process that occurs during the growth. As I see in WB and other figures, differentiation process is aborted, since there is only the initiation phase, characterized by the increase of FABP4, while maturation is almost absent since PPARg is non-existent according to WB. According to figure 2 with DAPI, with 10 and 20uM of ATV adipogenesis is almost non-existent, perhaps the low signal of immunostaining that the authors detected is unspecific. This suspicion is confirmed with the WB where neither PPARg nor adiponectin are present. Thus, according to the results obtained on HMSCs, what would occur in adipose tissue of the patients treated with ATV, is that there is no neoformation of adipocytes in subcutaneous adipose tissue.
Thank you very much for your thoughtful comments. The possibility that the reviewer was commented should be another explanation of our results.
-What is the effect of ATV on the mature adipocytes within the subcutaneous adipose tissue of these patients?
Unfortunately, our current study has been performed only with in vitro culture system, so that we have no detailed data to answer. Our future study will touch upon this question.
-My question is; whether the lipodystrophy detected with ATV treatment is, may be, not due to the inhibition of adipocytes neoformation, but rather to an imbalance between this process and the potential necrosis effects of ATV on mature adipocytes within the adipose tissue?.
At least the authors should discuss this issue
The authors do appreciate the comment, which is quite important to understand the process of adipose tissue injury. As far as we examined, apoptosis was the predominat cell death mode of adipocytes, while we rarely detected necrosis.
-In WB panels, it seems that the authors do not represent results from the same Blot as control profile seems to be different of that of the other proteins, for example; comparing beta-actin and FABP4 in figure 3.
Thank you very much for the comment. We have added an additional explanation about the protein control in the Materials and Methods section.
Reviewer 2 Report
I have only one concern, the authors should supply the results that the effects of ER stress inhibitors, GSK2606414 and 4μ8C, on the phosphorylation of their downstream (PERK and IRE-1), respectively.
Author Response
I have only one concern, the authors should supply the results that the effects of ER stress inhibitors, GSK2606414 and 4μ8C, on the phosphorylation of their downstream (PERK and IRE-1), respectively.
Thank you very much for your helpful comment, however, we unfortunately have those antibodies used for this purpose, so that it is unable to complete the response by due date.
The authors understand the importance of performing this point and will investigate in the next research protocol.
This manuscript is a resubmission of an earlier submission. The following is a list of the peer review reports and author responses from that submission.
Round 1
Reviewer 1 Report
The authors found that a protease inhibitor, ATV induces cell death and ER stress in differentiating adipocytes, but not adipose tissue-derived stem cells. While the findings are informative, stem cells are basically less sensitive to various stresses in comparison with differentiating or differentiated cells. The detailed effects of ATV on cells are not evaluated at all. Moreover, different effects on adipocytes between ATV and other protease inhibitors are not well discussed.
Adipocyte death is, of course, relevant to lipodystrophy, but the cell culture system only provided suggestion. There are several overstatements including the title in the manuscript.
In addition, this manuscript seems to have a lot of insufficient points. Examples are below.
Figure 3: “n” number is missing.
Figure 5: The focus of images are not good. I guess that DNA damage and apoptosis are not induced in ADSCs. Figure caption and legend are confusing.
Figure 6: Phosphorylated H2AX is an indicator of DNA damage, but not apoptotic cells. How did the author detect apoptotic cells?
Figure 4,7: It is not clear what Adipocytes D0 means. While neither FABP4 nor Adiponectin are expressed in Adipocyte D0 and ADSC D0, ATG12-ATG5 expression is different between Adipocyte D0 and ADSC D0. It is very confusing without any description. In addition, the relation between ER stress and adipocyte apoptosis should be examined in the author’s experimental settings.
Figure 8: CHOP expression is not clear. Much better images are necessary. “n” number is missing again.
Figure 9,10: Reasons that the authors examined ROS is not clearly described.
Table: related to Figure6, the method by which the authors calculated apoptotic cells is not described.
Reviewer 2 Report
- In line 55 and line 69, as there are reports showed PIs could induce ER stress and followed apoptosis, what is the creative for this manuscript? The authors should emphasize more details about the significance and creative of this manuscript.
- The authors need to combine the relative figures into a big figure, especially, such as the result and its related statistical analysis, figure 2 and figure 3.
- The authors should add more details about ATV treatments, such as when they add the ATV and how long the ATV treatment lasts, both for ADSCs and adipocyte under differentiation. Especially for adipocyte under differentiation, which may take 10 days or even longer with several medium changed, the authors need to add every details. Otherwise, it is hard to evaluate the results and conclusion.
- In line 97, please rectify the strange symbol.
- For figure 1A, please supply high quality images, also with high power images to show details.
- For figure 2C, 2D, and 4, depending on when the authors added the ATV, there should be two possibilities, one is ATV inhibited the adipocyte differentiation, another is that ATV inhibited the lipid droplet formation. The authors need to confirm which progress is influenced by ATV. And also, please also add high power images for figure 2.
- For Table 1, please indicate clearly for each column meaning.
- Please supply high quality images for figure 5.
- Please supply high quality images for figure 6. And please supply phosphorylated histone H2AX, DAPI and merge images for figure 6, both in low power and high power for evaluating the results and conclusion. The phosphorylated histone H2AX positive cells/total cells in the field should be calculated. And for these staining, the authors should also stain the cells with BODIPY or other markers for adipocyte to confirm these cells are adipocyte.
- For figure 7, what is the internal control? The ADSCs cultured for D5, D7, and D10 without ATV treatment should be also added for evaluating the effect of ATV on ADSCs. Although the statistical analysis should be added to each western blot result, there are several special ratio should be addressed, eg, Cleaved casepase-3/ pro-casepase-3, LC3B-II/LC3B-I.
- Please supply high quality images for figure 8 and 9. And please supply antibody staining, DAPI and merge images for figure 8 and 9, both in low power and high power for evaluating the results and conclusion. The positive cells/total cells in the field should be calculated. And for these staining, the authors should also stain the cells with BODIPY or other markers for adipocyte to confirm these cells are adipocyte.
- The statistical analysis should be added to each western blot result. The authors should describe how many repeats for each experiment.
- The authors should add the inhibitors of ER stress or specific siRNA targeted to ER stress to inhibiting ER stress, if these could reverse the effect of ATV, the results will support their assumption. The data showed here is not enough to support their conclusion.
- The authors linked their findings with lipodystrophy, however, they haven’t shown any animal experiments. They should use ADSCs and adipocyte from lipodystrophy human patient or animal model to confirm the findings they revealed here. And for their therapeutic purpose, they also should inject the lipodystrophy animals with inhibitors of ER stress or AAV mediated shRNA targeted to ER stress, to check whether there will be a relief phenotype for those animals.